# PICH impacts the spindle assembly checkpoint via its DNA translocase and SUMO-interaction activities

Bunu Lama[1], Hyewon Park[1], Anita Saraf[2], Victoria Hassebroek[1], Daniel Keifenheim[3], Tomoko Saito-Fujita[4], Noriko Saitoh[4] ⓘ, Vasilisa Aksenova[5], Alexei Arnaoutov[5], Mary Dasso[5] ⓘ, Duncan J Clarke[3] ⓘ, Yoshiaki Azuma[1] ⓘ

**Either inhibiting or stabilizing SUMOylation in mitosis causes defects in chromosome segregation, suggesting that dynamic mitotic SUMOylation of proteins is critical to maintain integrity of the genome. Polo-like kinase 1–interacting checkpoint helicase (PICH), a mitotic chromatin remodeling enzyme, interacts with SUMOylated chromosomal proteins via three SUMO-interacting motifs (SIMs) to control their association with chromosomes. Using cell lines with conditional PICH depletion/PICH replacement, we revealed mitotic defects associated with compromised PICH functions toward SUMOylated chromosomal proteins. Defects in either remodeling activity or SIMs of PICH delayed mitotic progression caused by activation of the spindle assembly checkpoint (SAC) indicated by extended duration of Mad1 foci at centromeres. Proteomics analysis of chromosomal SUMOylated proteins whose abundance is controlled by PICH activity identified candidate proteins to explain the SAC activation phenotype. Among the identified candidates, Bub1 kinetochore abundance is increased upon loss of PICH. Our results demonstrated a novel relationship between PICH and the SAC, where PICH directly or indirectly affects Bub1 association at the kinetochore and impacts SAC activity to control mitosis.**

## Introduction

During mitosis, a mammalian cell must segregate its duplicated chromosomes into two daughter cells evenly, which requires spatiotemporal coordination of each event in mitosis, including structural organization of mitotic chromosomes, untangling interwind genomic DNA, and establishment of bipolar spindle attachment of kinetochores (Paulson et al, 2021). Cells possess monitoring mechanisms for this coordination and can control the timing of the mitotic events to avoid chromosome segregation errors that result in genome instability. The spindle assembly checkpoint (SAC) monitors the bipolar attachment of kinetochores to the mitotic spindle (Lara-Gonzalez et al, 2021), and the topo-isomerase II–responsive checkpoint (TRC) monitors entangled genomic DNA/stalled detangling activity (Soliman et al, 2023). Various post-translational protein modifications (PTMs) are critical for these checkpoints to function.

The PTM by SUMO (small ubiquitin-related modifier), SUMOylation, is a dynamic, reversible process that occurs by covalent post-translational modification of specific lysine (K) residues on target proteins via enzymatic cascade reactions similar to ubiquitination. Since its discovery as a suppressor of Mif2 mutation in budding yeast, mitotic defects in SUMOylation pathway mutants are commonly observed in many model organisms with various phenotypes (Mukhopadhyay & Dasso, 2017). The dynamics of SUMOylation in mitosis plays a key role in robust mitosis as both inhibition of SUMOylation and inhibition of deSUMOylation result in defects in mitotic events (Azuma et al, 2003; Cubenas-Potts et al, 2013; Pelisch et al, 2014). Supporting the pleiotropic role of SUMOylation in mitotic events, proteomics analysis identified several proteins that can explain the phenotypes (Denison et al, 2005; Schou et al, 2014; Cubenas-Potts et al, 2015), which includes SUMOylation of TopoIIα needed for TRC activation (Edgerton et al, 2016; Yoshida et al, 2016; Pandey et al, 2020) and SUMOylation of centromere/kinetochore proteins for efficient kinetochore assembly (Zhang et al, 2008; Subramonian et al, 2021). The function of SUMOylation in these cases is promoting protein/protein interaction via SUMO and SUMO-interacting motifs (SIMs), which can help in the recruitment of key mitotic enzymes to SUMOylated proteins and control protein complex formation at specific chromosomal locations. The finding that PICH (Polo-like kinase 1–interacting checkpoint helicase) is a SUMOylated protein–interacting DNA translocase suggests an additional role of SUMOylation in chromosomal proteins.

PICH is a DNA translocase belonging to the SNF2 family that can bind DNA under tension and remodel chromatin in vitro (Baumann et al, 2007; Biebricher et al, 2013; Spakman et al, 2022). That activity

[1]Department of Molecular Biosciences, University of Kansas, Lawrence, KS, USA   [2]Mass Spectrometry and Analytical Proteomics Laboratory, University of Kansas, Lawrence, KS, USA   [3]Department of Genetics, Cell Biology and Development, University of Minnesota, Minneapolis, MN, USA   [4]Division of Cancer Biology, The Cancer Institute of Japanese Foundation for Cancer Research, Tokyo, Japan   [5]Division of Molecular and Cellular Biology, National Institute for Child Health and Human Development, National Institutes of Health, Bethesda, MD, USA

Correspondence: azumay@ku.edu
Victoria Hassebroek's present address is Genome Engineering Department, Stowers Institute for Medical Research, Kansas City, MO, USA

 

could provide a mechanism to explain how PICH acts on remaining tangled genomic DNA in anaphase, ultrafine DNA bridges (UFBs), where PICH is specifically localized. In addition to that specific localization and potential role in resolving UFBs, loss of PICH function results in chromosome morphological defects in mitosis, including improper chromosome structural organization and impaired resolution of sister chromatids, as well as micronucleus formation (Kurasawa & Yu-Lee, 2010; Kaulich et al, 2012; Nielsen et al, 2015). These cellular levels of mitosis defects suggest that PICH could have additional functions beyond the resolution of UFBs in anaphase. PICH possesses three SIMs, and it interacts with SUMOylated proteins promiscuously (Sridharan et al, 2015; Sridharan & Azuma, 2016). Because DNA translocase activity-dead mutants of PICH strongly interact with SUMOylated proteins on chromosomes, PICH's remodeling activity could control the association of chromosomal SUMOylated proteins to regulate their dynamics on chromosomes (Hassebroek et al, 2020). But why SUMOylated proteins need to be remodeled on mitotic chromosomes by PICH and the identity of those SUMOylated proteins remains unanswered.

To explore the functional relationship between PICH and SUMOylated proteins in mitosis, we quantified mitotic progression after conditional PICH knockdown using the auxin-inducible degron system and tetracycline-induced exogenous PICH expression in CRISPR/Cas9 genome-edited cell lines (Hassebroek et al, 2020). The results reveal that PICH remodeling of SUMOylated proteins contributes to the proper temporal silencing of the SAC. Depletion of PICH or the expression of mutants of PICH that lack SUMO binding ability (PICH d3SIM) or translocase activity (PICH K128A) causes delay in mitosis because of prolonged activation of the SAC. SUMOylated chromosomal proteomics identified several potential SUMOylated chromosomal proteins controlled by PICH. Among them, we validated Bub1 as being a SAC protein with increased chromosomal abundance upon PICH loss. The results suggest Bub1 is a target of PICH remodeling activity that directly or indirectly impacts SAC silencing.

# Results and Discussion

## Both SUMO-interacting ability and translocase activity of PICH are required for normal mitotic progression

Failure in the regulation of mitotic SUMOylation resulted in a defect in the progression of mitosis (Seufert et al, 1995; Myatt et al, 2014; Eifler et al, 2018). Because PICH can control mitotic chromosomal SUMOylated proteins, we sought to examine whether the loss of PICH function causes defective progression through mitosis using conditional PICH depletion/PICH replacement to mutants (Fig 1A) using AID-mediated degradation and Tet-inducible expression (Hassebroek et al, 2020). Toward that goal, we tagged endogenous H2B with miRFP680 in conditional PICH depletion/replacement lines (Figs 1B and S1A) and then measured the duration of mitosis by live-cell imaging of the miRFP680 signal. To obtain optimal PICH expression and the number of mitotic cells, we synchronized cells using a single thymidine arrest/

release procedure combined with Aux/Dox treatment, as indicated in Fig 1C. Live imaging showed that AID-mediated PICH depletion extended the duration of mitosis (Fig 1D and E). The mean duration of mitosis increased by 15% in the PICH-depleted condition compared with the control, which had a mean duration of 32.7 min. Specifically, the period from chromosome condensation, represented by distribution changes of the H2B signal, to anaphase onset, when we observed clearly separated chromatids, was increased (Fig 1D). Although the depletion of PICH did not cause arrest in mitosis, PICH depletion significantly increases the duration of mitosis. Unlike results from other studies (Leng et al, 2008; Kurasawa & Yu-Lee, 2010; Kaulich et al, 2012; Nielsen et al, 2015), which showed that depletion of PICH did not affect the timing of mitotic progression, we observed an increase in mitotic duration upon loss of PICH. But Nielsen's report showed an increase in the doubling time of cells upon PICH knockdown, and cell cycle distribution showed more cells in the G2/M phase, which supports our observation of mitotic delay upon PICH depletion. The extended mitosis we observed was rescued by PICH WT expression (PICH WT) but not by the ATPase-deficient mutant (PICH-K128A) or a non–SUMO-interacting form of PICH (PICH d3SIM) (Fig 1D and E). In PICH-K128A and PICH-d3SIM conditions, the mean duration of mitosis increased by 20% compared with the control. The results suggest that both translocase activity and the SUMO-interacting ability of PICH are required to support the normal progression of mitosis.

## The spindle attachment checkpoint is activated upon loss of PICH function

Considering that delay in mitosis usually indicates checkpoint activation, we sought to determine which checkpoint contributes to mitotic delay in the loss of PICH function mutants. Because SUMOylated topoisomerase II alpha can mediate mitotic delay upon inhibition of TopoII activity (TRC) (Pandey et al, 2020) and PICH regulates SUMOylated TopoIIα abundance on the mitotic chromosome (Hassebroek et al, 2020), we hypothesized that loss of PICH function might activate the TRC and thus extend the duration of mitosis by increasing SUMOylated TopoIIα on mitotic chromosomes. To test this, we created double AID-3XFLAG cell lines with PICH-AID and TopoIIα-AID and replaced TopoIIα with either TopoIIα WT or TopoIIα 3KR where all three C-terminal domain lysine residues are mutated to arginine, which inhibits SUMO2/3 conjugation on the CTD (Ryu et al, 2015; Yoshida et al, 2016; Pandey et al, 2020). If the mitotic delay upon PICH depletion was due to TRC activation, the cells with TopoIIα-3KR should progress normally through mitosis even without PICH function. Western blotting confirmed the codepletion of PICH and TopoIIα with Aux treatment and exogenous TopoIIα expression after Dox addition with the consistent cell synchronization culture condition as in Fig 1C (Fig 2A). Duration of mitosis was measured by following the mCherry-TopoIIα signal (Fig 2B), with the same criteria as in Fig 1, measuring the duration from chromosome condensation and separation of sister chromatids represented by distribution of the mCherry-TopoIIα signal.

Contrary to our prediction, cells took even longer to complete mitosis upon the expression of TopoIIα-3KR, suggesting that the

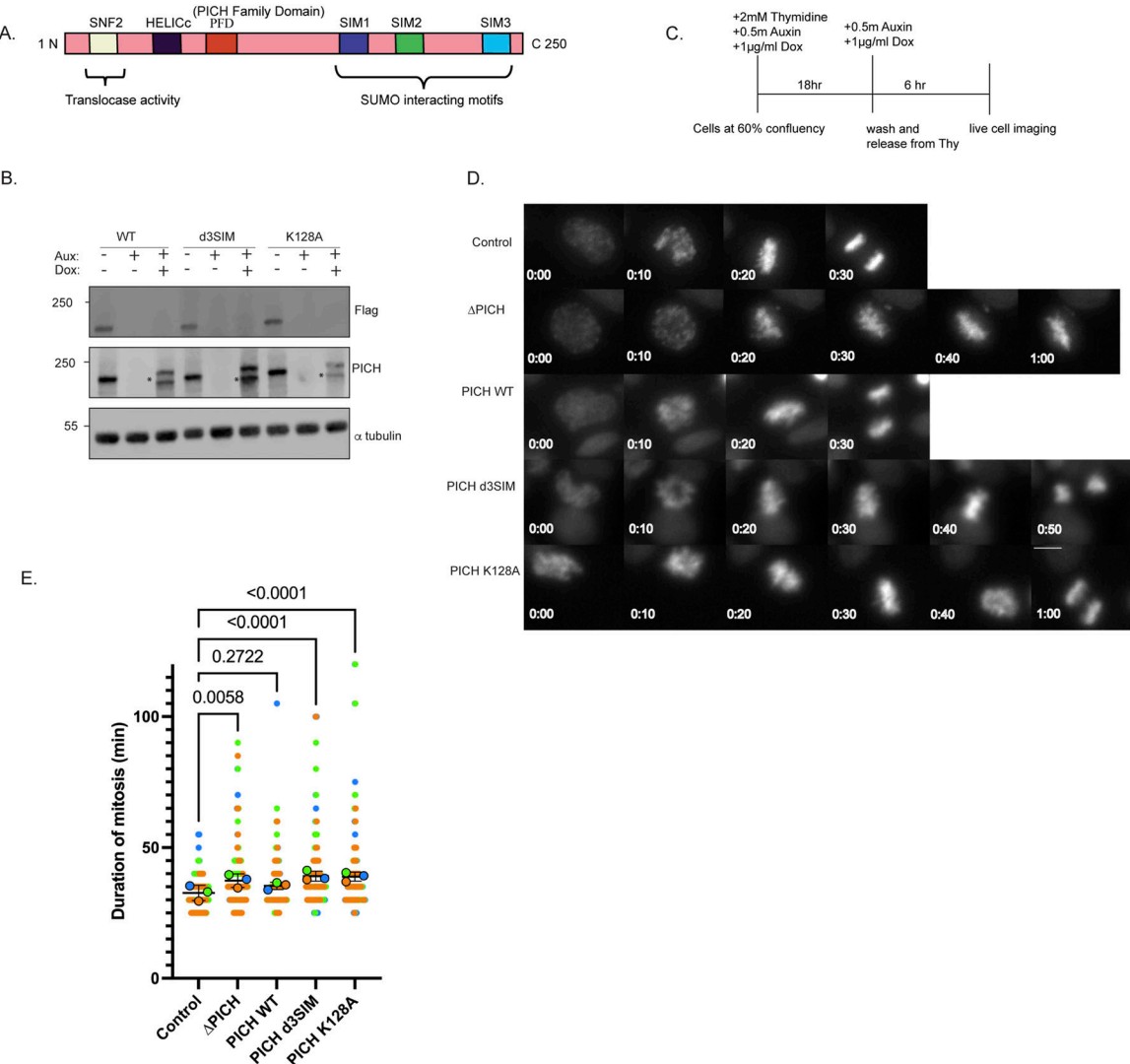

**Figure 1. Depletion of PICH leads to an increase in mitotic timing, and both SUMO-interacting motifs and translocase activity of PICH are required for normal mitotic progression.**
**(A)** Schematic diagram of PICH showing its three SUMO-interacting motifs and its domain for translocase activity. **(B)** DLD-1 cells with endogenous H2B tagged with miRFP680 in a PICH replacement background cell line were treated with auxin and doxycycline for 22 h, and whole-cell lysates were subjected to Western blot with indicated antibodies. * indicates degraded exogenous mCherry-PICH. **(C)** Scheme for cell synchronization for live-cell imaging. **(D)** Live-cell imaging of cells undergoing mitosis captured from nuclear envelope breakdown to anaphase with the H2B-miRFP680 signal. mCherry-positive cells were counted for PICH replacement cells. Scale bar = 10 μm. **(E)** Quantification of the timing of cells in mitosis. n = 3 experiments with at least 50 cells counted per condition. *P*-values indicate one-way ANOVA followed by Tukey's multiple comparison correction. Horizontal bars indicate the mean, and error bars indicate the SD.
Source data are available for this figure.

activation of TRC does not contribute to the loss of PICH-mediated delay (Fig 2B and C). Notably, simply replacing TopoIIα with the TopoIIα-3KR mutant (without PICH depletion) does not change the duration of mitosis (Fig S2A). Although the genetic backgrounds of the double AID line in Fig 2 and those of the single TopoIIα-replaced line in Fig S2 are distinct, the results indicate that the lack of SUMOylation on TopoIIα-CTD does not affect mitosis duration. The initial step of TRC activation involves TopoIIα SUMOylation at the C-terminal domain, which regulates the localization of the histone H3 kinase Haspin at the centromere and recruitment of chromosomal passenger complexes (CPC) to kinetochores in mitosis

(Yoshida et al, 2016). Therefore, to further validate our findings, we examined Haspin localization under PICH depletion conditions. We engineered an endogenous Haspin-mNeon cell line in the PICH-AID background (Fig S1B) and performed live-cell imaging upon PICH depletion to analyze Haspin localization on mitotic chromosomes (Fig S2B). Collected images from each condition were applied to a machine learning algorithm, wndchrm (Shamir et al, 2008), as we performed previously for TopoIIα mutant localization on mitotic chromosomes (Sundararajan et al, 2023). The wndchrm analysis comparing the control condition and PICH-depleted condition showed an accuracy percentage of 50%, implying that the PICH

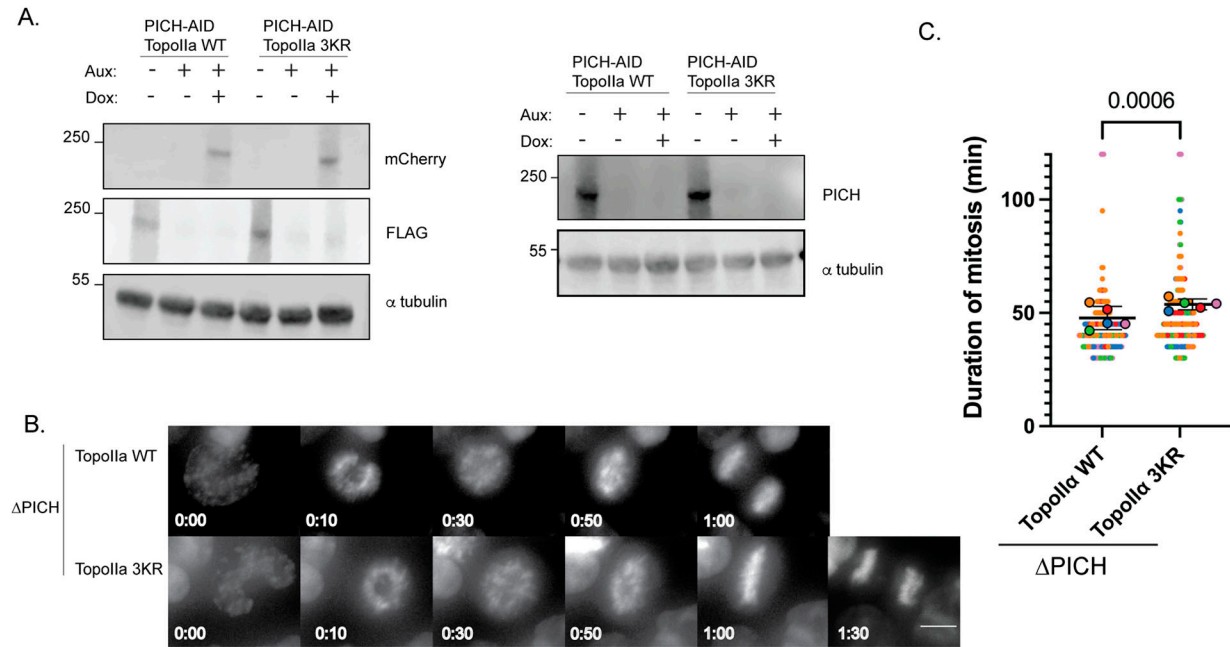

**Figure 2. Delay in mitosis upon PICH depletion is not contributed to the topoisomerase IIα–responsive checkpoint.**
**(A)** DLD-1 cells with endogenous PICH and TopoIIα tagged with AID and replaced with TopoIIα or TopoIIα-3KR were treated with auxin and doxycycline for 22 h, and the whole-cell lysate was subjected to Western blot with indicated antibodies. **(B)** Live-cell imaging of cells undergoing mitosis captured from nuclear envelope breakdown to anaphase with the mCherry signal. Scale bar = 10 μm. **(C)** Quantification of the timing of cells in mitosis. n = 5 experiments with at least 30 cells counted per condition. *P*-value indicates a two-tailed unpaired-sample *t* test. The horizontal bar indicates the mean, and the error bar indicates the SD.
Source data are available for this figure.

depletion does not change mitotic chromosomal Haspin localization (Fig S2C). Together, this suggests that TRC was not activated upon PICH depletion.

We found that TRC activation upon PICH depletion does not contribute to PICH-dependent mitotic delay. Next, we sought to determine whether the SAC is activated upon loss of PICH. Thus, we examined Mad1 retention time on the kinetochore to indicate the duration of SAC activation (Shah et al, 2004). To assess this, endogenous Mad1 was tagged with mNeon in the PICH-AID cell line (Fig S1C) and we performed live-cell imaging to determine Mad1 retention time on the kinetochores after nuclear envelope breakdown (NEBD) with loss of PICH (Fig 3A and B). Mad1 focus duration represents the time from the disappearance of nuclear envelope–localized Mad1 (onset of NEBD) to the disappearance of kinetochore Mad1 foci after completion of metaphase alignment. Mad1 focus duration was extended by PICH depletion, indicating that loss of PICH activates the SAC (Fig 3B and C). Because both the loss of SUMO-interacting ability and translocase activity of PICH also showed a delay in mitosis, we examined whether the loss of these functions of PICH also activates the SAC. Analysis of Mad1 focus duration with replacement of PICH to PICH d3SIM and PICH K128A revealed that Mad1 foci are retained on kinetochores for a longer time, suggesting that PICH contributes to SAC silencing to promote normal mitotic progression using its remodeling activity toward the SUMOylated chromosomal protein (Fig 3D).

## Identification of chromosomal SUMOylated proteins controlled by PICH reveals several candidate targets of PICH remodeling activity

Our data so far support that the SUMO-interacting motifs and translocase activity of PICH cooperate to ensure normal mitotic progression by controlling the SAC, suggesting that SUMOylated chromosomal proteins remodeled by PICH are involved. Because SUMOylated TopoIIα, previously shown to be controlled by PICH (Hassebroek et al, 2020), is not responsible for the PICH-induced mitotic delay, we wanted to ask which chromosomal proteins are responsible. Toward this goal, we created cell lines in which endogenous SUMO2 is tagged by hexahistidine (Figs 4A and S1D). Polyhistidine-tagged SUMO has been employed to enable the isolation of SUMOylated proteins under denaturing conditions that abolish deSUMOylation by SENPs (Vertegaal et al, 2004, 2006). To identify chromosomal SUMO2-modified proteins remodeled by PICH in mitosis, we created cell lines with PICH depletion/replacement to the PICH-K128A mutant (Fig 4A) and applied an established purification method for proteome analysis by mass spectrometry (Fig 4B) (Cubenas-Potts et al, 2015). We expected the loss of PICH to stabilize SUMO2/3-modified proteins on chromosomes. Thus, critical PICH targets that are SUMOylated on chromosomes would have increased abundance in mass spectrometry analysis when PICH function is compromised. Because the PICH K128A mutant showed increased colocalization with chromosomal SUMOylated proteins (Hassebroek et al, 2020), we expected this

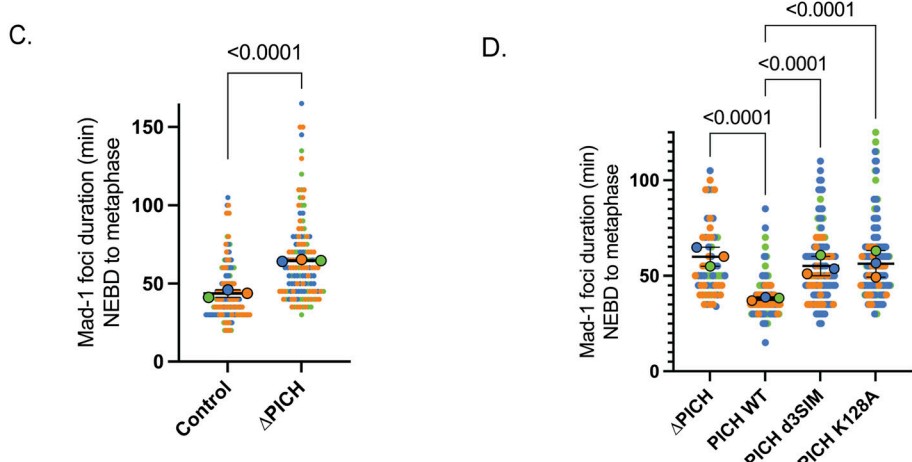

**Figure 3.  PICH depletion increases Mad1 focus duration on kinetochores indicating spindle assembly checkpoint activation.**
**(A)** Scheme of experimental condition for live-cell imaging. **(B)** Mad1-mNeon focus duration was counted from nuclear envelope breakdown to metaphase from asynchronous cells in the ΔPICH condition. Scale bar = 5 μm. **(C)** Quantification of Mad1 focus duration in control versus ΔPICH. *P*-values for comparison from three replicates were calculated using two-tailed unpaired-sample *t* tests. **(D)** Quantification of Mad1-mNeon focus duration was counted from nuclear envelope breakdown to metaphase from asynchronous cells in the PICH-replaced condition, n = 3 experiments. *P*-values indicate one-way ANOVA followed by Tukey's multiple comparison correction. Horizontal bars indicate the mean, and error bars indicate the SD.

mutant to further stabilize specific SUMOylated chromosomal proteins targeted by PICH remodeling activity.

Isolated mitotic chromosomal fractions from these samples showed the expected increase of SUMOylated proteins with PICH depletion and PICH-K128A replacement (Fig 4C). After isolation of the SUMOylated fractions (Fig 4D), mass spectrometry analysis was performed for three experimental replicates. Normalized spectral abundance factor (NSAF), which represents the relative abundance of proteins in the samples, was calculated from the total number of spectra (SpC) used for identification of each protein and the number of the amino acids of the proteins (L) (Florens et al, 2006; Paoletti et al, 2006). The NSAF ratio was calculated between control and experimental conditions, PICH depletion or PICH K128A re-placement, and the ratio of the NSAF value was used for estimating the relative abundance of identified proteins in samples (Sda-taF4.1). Initially, the proteins detected at least twice in one of the three conditions were selected (SdataF4.2, selected protein list tab). A total of 1,662 proteins were within the initial selection criteria, and among the selected proteins, there were known chromosomal structural proteins, centromeric/kinetochore components, and checkpoint mediators (SdataF4.2, short protein list tab). Changes in the NSAF of these proteins were summarized in a heat map, and most of these showed increased abundance when PICH was

depleted or replaced with PICH-K128A (Fig S3). Because the mitosis delay phenotype is conserved between PICH depletion and PICH K128A replacement conditions, we further analyzed identified proteins that repeatedly showed consistent differences in the NSAF ratio in the experimental conditions over control. The PICH-depleted sample and PICH-K128A–replaced sample showed 215 proteins, and 149 proteins had increased abundance (SdataF4.3). Among them, 106 proteins are commonly increased (Fig 4E). As a proof of principle for this analysis, we observed increased TopoIIα abundance in both the PICH-depleted sample and the PICH-K128A–replaced sample (Fig 4F), as shown previously (Hassebroek et al, 2020). Among the 106 candidates, 17 proteins that have roles in mitosis are shown in the heat map (Fig 4F), and some of them could contribute to the mitotic delay phenotype observed with loss of PICH function, such as Bub1, a SAC protein, and other centromere/kinetochore proteins.

### PICH attenuates the kinetochore association of Bub1

Among the candidate proteins identified from the mass spectrometry, we focused on Bub1 as it is a key regulator of the SAC. If the loss of PICH function increases or extends Bub1 activity toward Mad1 at kinetochores, it can explain the prolonged activation of the

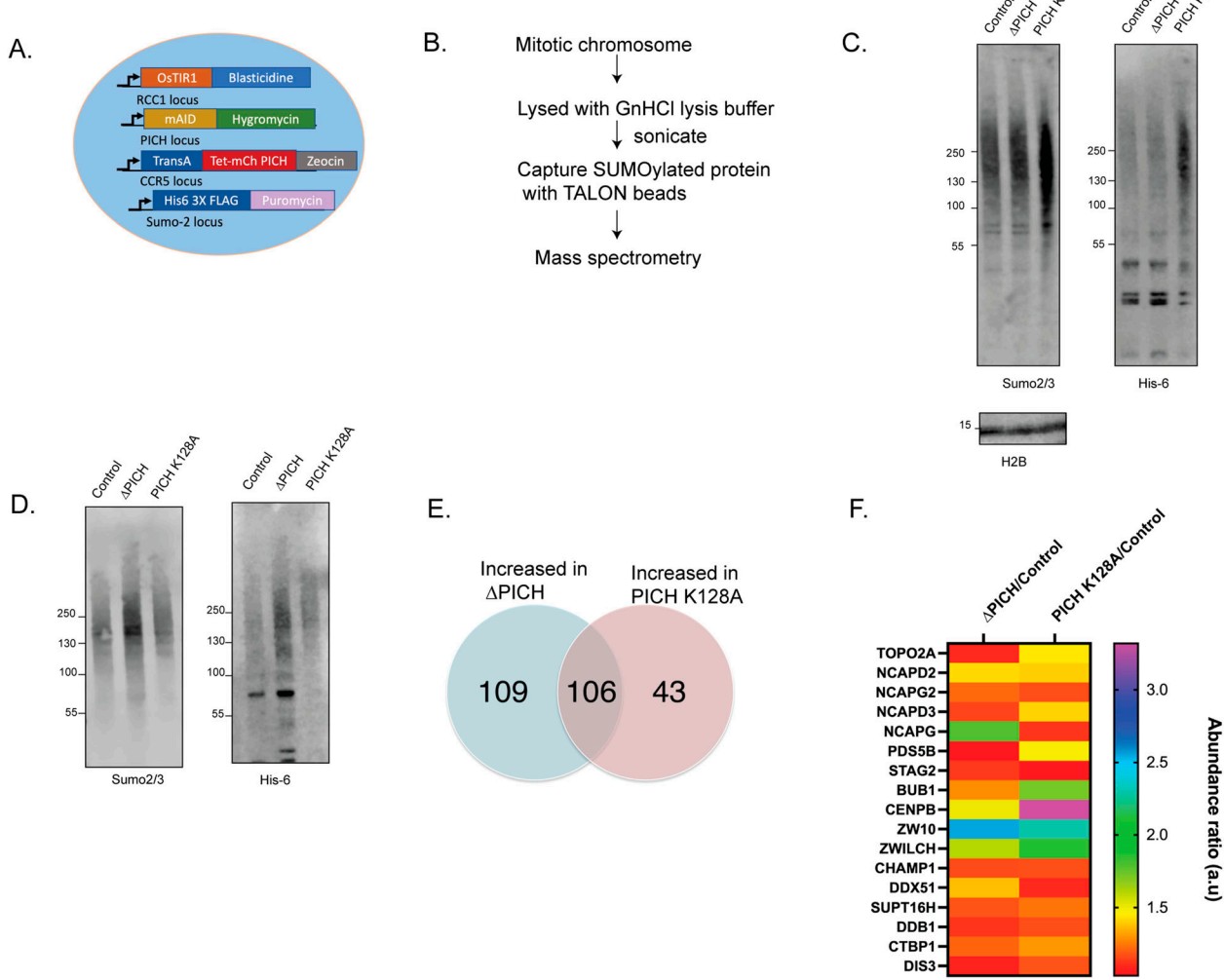

**Figure 4. Mass spectrometry reveals PICH-interacting candidate proteins.**
**(A)** Schematic representation of DLD-1 cells with endogenous Sumo-2 tagged with histidine 6-3XFLAG in PICH-AID and PICH replacement cell lines. **(B)** Workflow of SUMOylated protein isolation under the denatured condition followed by mass spectrometry. **(C)** Western blot of mitotic chromosome sample showing SUMO2/3 and His6 signals. **(D)** Western blot of purified fraction after SUMOylated protein isolation probed with antibodies against SUMO2/3 and His-6. **(E)** Venn diagram showing the number of proteins identified from mass spectrometry that is increased in each condition. **(F)** Heat map showing potential PICH-interacting proteins.
Source data are available for this figure.

SAC after loss of PICH function. To examine that possibility, endogenous Bub1 was fused with mNeon in a cell line that has endogenous CENP-A tagged with miRFP680 and AID-fused PICH for conditional PICH depletion (Fig S1E and F). Mitotic chromosomes from synchronized cell cultures were imaged for quantification of Bub1 located on the kinetochore, in the vicinity of CENP-A. By measuring the signal intensities, we observed increased levels of Bub1 in PICH-depleted cells as compared to the control (Fig 5A and B). Notably, CENP-A signals did not show a detectable difference with PICH depletion, suggesting centromeric chromatin organization might not be affected by the loss of PICH function (Fig 5C). Measuring Bub1 focus duration after release from colcemid-induced arrest revealed an extended presence of Bub1 foci at centromeres after PICH was depleted (Fig 5D and E). Prolonged Bub1 foci after release from colcemid-induced SAC activation indicate

that PICH function is necessary for efficiently relocating Bub1 from the kinetochore. This may involve either directly relocating Bub1 or regulating the efficiency of microtubule attachment to the kinetochore (Fig 5F).

SUMOylation of Bub1 was reported during meiotic chromosome segregation (Pelisch et al, 2019). If Bub1 is SUMOylated in mitotic chromosomes, it could be targeted by PICH via the SIMs and then remodeled by PICH translocase activity. Because Bub1 initiates SAC activation (Sharp-Baker & Chen, 2001; Moyle et al, 2014), it is possible that increased Bub1 upon loss of PICH could contribute to the activation of SAC and prolonged Mad1 focus duration (Fig 3). Bub1 abundance and duration at centromeric regions were increased upon loss of PICH function, suggesting that simply increasing the Bub1 amount could activate Mad1-mediated SAC. To further verify this model, detection of SUMOylated Bub1 and/or

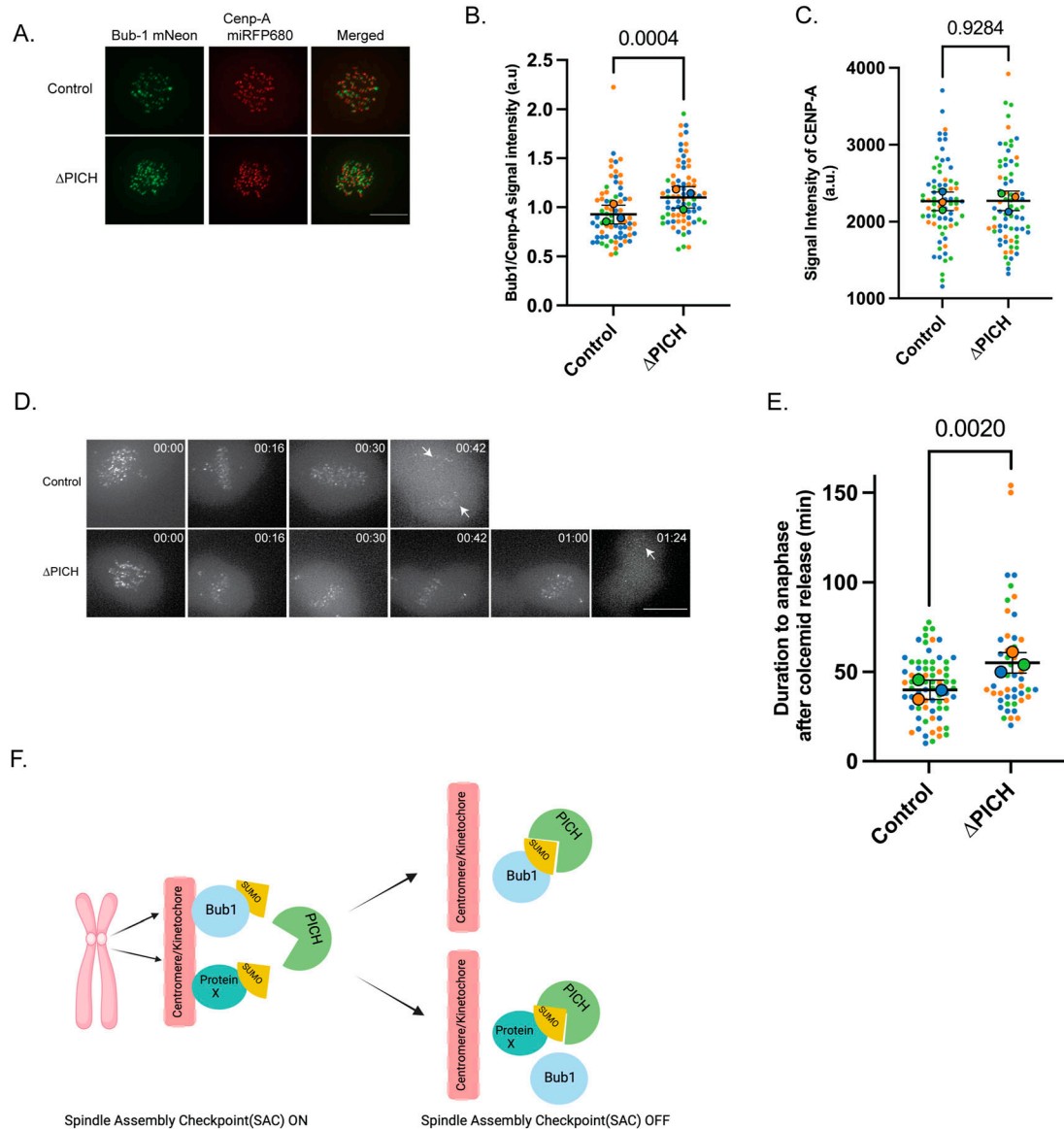

**Figure 5. Bub1 levels on chromosomes are regulated by PICH.**
**(A)** DLD-1 cells with endogenous Bub1 tagged with mNeon and CENP-A tagged with miRFP680 were synchronized with thymidine and imaged 6 h after release. Scale bar = 8 μm. **(B)** Signal intensities (a.u) of Bub1-mNeon normalized by CENP-A are shown with the mean value and SD. P-values for comparison from three replicates were calculated using two-tailed unpaired-sample t tests. **(C)** Signal intensities (a.u) of CENP-A in control and PICH-depleted conditions. P-values for comparison from three replicates were calculated using two-tailed unpaired-sample t tests. **(D)** Cells were released from colcemid arrest and imaged until anaphase onset tracking the Bub1-mNeon signal. The arrow shows Bub1-mNeon foci in anaphase. Scale bar = 10 μm. **(E)** Quantification of duration to anaphase after colcemid release. n = 3. The P-value was calculated using two-tailed unpaired-sample t tests. **(F)** Proposed model of how PICH impacts spindle assembly checkpoint. PICH either removes SUMOylated Bub1 from the chromosomes or remodels other SUMOylated proteins, which in turn promotes dissociation of Bub1 to inactivate the spindle assembly checkpoint, and cells can progress to anaphase.

analysis of dynamics of Bub1 at kinetochores upon loss of PICH function will be the next approach. Although direct control of SUMOylated Bub1 by PICH is a possible mechanism, there are other feasible candidate proteins that could explain the mitosis delay phenotype in our candidate protein list (Supplemental Data 3 and Fig 4). Other possibilities we can see from our candidate list include the following: (1) PICH controls the efficiency of microtubule attachment to kinetochores via the RZZ complex (ZW10 and Zwelch) (Buffin et al, 2005; Kops et al, 2005) or CHAMP1 (Itoh et al, 2011), and

(2) PICH controls centromeric structure via CENP-B (Nagpal & Fierz, 2021; Chardon et al, 2022) and/or chromatin regulators (SUPT16H) (Prendergast et al, 2016). Although we did not detect a difference in CENP-A signals (Fig 5C), there might be a possibility that PICH controls centromeric chromatin or centromere structure via chromatin regulators and CENP-B. The next challenge will be confirmation of SUMOylation of these candidates, then determining the SUMOylation sites for these candidates, and then testing whether non-SUMOylatable mutants of the candidates affect the

mitotic progression defect after loss of PICH as we performed with TopoIIα-3KR here (Fig 2).

We also note that condensins were identified as PICH-targeted SUMOylated chromosomal proteins (Fig 4F). Because condensin SUMOylation has been previously reported, it is possible that PICH can directly regulate these chromosome structural proteins using its translocase activity. Condensation defects are phenotypes observed after loss of PICH function (Nielsen et al, 2015); thus, it is intriguing to speculate that PICH controls the SUMOylated condensin pool of proteins to promote proper condensation of the mitotic chromosomes. The condensation problem could originate from the misregulation of condensin on the chromosomal arm regions, which could be distinct from SAC regulation at the kinetochore. Loss of PICH also causes increased chromosome bridge formation, and SUMO-interacting ability of PICH contributes to preventing chromosome bridge formation. Intriguingly, we observed different contributions of the SIMs of PICH for suppressing chromosome bridge formation versus centromeric localization of PICH. Specifically, SIM1 and SIM2 prevent chromosome bridge formation and SIM3 is needed for centromeric localization of PICH (Sridharan & Azuma, 2016). Further proteomics studies using PICH-specific mutants will help to dissect PICH's array of functions on mitotic chromosomes, that is, SAC regulation at the kinetochore and structural organization of chromosomes at arm regions.

Overall, our findings indicate that conditional depletion of PICH results in misregulation of the SAC, thereby extending the duration of mitosis. PICH is involved in remodeling SUMOylated proteins on the chromosomes, which in turn regulates their levels. Therefore, depletion of PICH impairs SUMO protein remodeling, leading to an accumulation of SUMOylated proteins on chromosomes, and negatively affects mitosis. The findings presented here have implications for understanding the addiction of cancer cells to PICH and the mechanism leading to their death when PICH is depleted (Huang et al, 2019; Xie et al, 2019; Chen et al, 2021). Our results indicate that PICH typically silences the SAC by remodeling SUMO proteins. One explanation why cancer cells are addicted to PICH may be that overexpressed PICH is needed to silence the SAC for rapid division. Yang et al (2023) have shown that depletion of PICH in breast cancer cells induces cell cycle arrest with the accumulation of DNA bridges leading to chromosome instability (Yang et al, 2023). Therefore, depleting PICH in the cancer cells activates the SAC, inhibiting cell proliferation and inducing apoptosis.

# Materials and Methods

## Plasmid and DNA constructs

### DNA construct and antibody preparation

The donor plasmid construction and configuration for *OsTIR1* integration, AID tagging to TopoIIα, AID tagging to PICH, and inducible expression cassette targeting to *CCR5* or *hH11* locus were described previously (Hassebroek et al, 2020). For double AID tagging to TopoIIα and PICH for Fig 2, the AID-PICH tagging donor was modified for converting the hygromycin-resistant gene to the puromycin registrant gene. Also, the *hH11* TopoIIα expression cassette plasmid

was modified to replace its antibiotics resistant from puromycin to Zeocin. In this study, the donor plasmids were created for miRFP680 fusion to endogenous CENP-A and H2B at their C terminus, mNeon tagging to Mad1, Bub1, and Haspin, and hexahistidine with 3xFLAG (His6-3xFLAG) tagging to SUMO2. In all cases, homology arm DNA fragments were isolated using primers listed in the table (Table S1) using either the genomic DNA or the obtained donor plasmid as a template. For the H2B donor, the original H2B C-terminal tagging plasmid, AICSDP-52: HIST1H2BJ-mEGFP, was a gift from Allen Institute for Cell Science (plasmid # 109121; Addgene; http://n2t.net/addgene:109121; RRID:Addgene_109121). For mNeon fusion donors, amplified homology arms were used to replace homology arm sequences of the mNeon-PICH donor (Sundararajan et al, 2023). For the miRFP680 tagging donor, the miRFP680 DNA fragment was isolated from the original plasmid, pmiRFP680-N1 (Matlashov et al, 2020), by PCR amplification. pmiRFP680-N1 was a gift from Vladislav Verkhusha (plasmid # 136557; Addgene; http://n2t.net/addgene:136557; RRID:Addgene_136557). The CENP-A miRFP680 donor plasmid was created by swapping the miRFP670 sequence of the CENP-A C-terminal miRFP670 fusion donor plasmid (Sundararajan et al, 2023) to the miRFP680 DNA fragment. Then, CENP-A homology arms were replaced with H2B homology arms to create the H2B-miRFP680 donor plasmid. For His6-3xFLAG tagging, fusion PCR was performed to create His6-3xFLAG fragment with GA linker using primers indicated in Table S1. The pET30 plasmid and AID-3xFLAG donor plasmid were used as a template to amplify the His6-3xFLAG fusion fragment. The His6-3xFLAG fragment was swapped with the mNeon fragment in N-terminal mNeon tagging for PICH (Sundararajan et al, 2023). Then, homology arms were replaced with SUMO2 homology arms to create the donor plasmid of His6-3xFLAG tagging to endogenous SUMO2. The guide RNA sequences summarized in Table S1 were designed using CRISPR design tools from Zhang laboratory, MIT or CRISPOR (Concordet & Haeussler, 2018). The synthesized oligo DNA primers for the guides were inserted into pX330 (plasmid #42230; Addgene: https://www.addgene.org/42230/; RRID:Addgene_42230) or pX330A-2 (plasmid #58766; Addgene; https://www.addgene.org/58766/; RRID:Addgene_58766). All constructs were verified by DNA sequencing.

The anti-PICH and anti-SUMO2/3 antibodies were generated as previously described (Ryu et al, 2010; Hassebroek et al, 2020). Other commercial antibodies used in this study are listed in Table S2.

## Cell culture, transfections, and colony isolation

Targeted integration in the genome was done using the CRISPR/Cas9 system as previously described (Hassebroek et al, 2020). DLD-1 cells were transfected with the donor plasmid and the guide plasmid using ViaFect (#E4981; Promega) on 3.5-cm dishes. After 2 d of transfection, the cells were split into 10-cm dishes, making them about 20% confluent. From the next day, they were subjected to a selection process by maintaining them in a desired selective antibiotic (1 mg/ml blasticidin [#ant-bl; InvivoGen], 0.5 mg/ml puromycin [#ant-pr; InvivoGen], 200 mg/ml hygromycin B gold [#ant-hg; InvivoGen]). The cells were cultured for 10–14 d, and then, the colonies were picked and plated on a 48-well plate. The clones were subjected to genomic PCR and Western blot to verify transgene insertion. Genomic DNA was

isolated by pelleting cells and lysis using lysis buffer (100 mM Tris–HCl [pH 8.0], 200 mM NaCl, 5 mM EDTA, 1% SDS, and 0.6 mg/ml proteinase K [#P8107S; NEB]) followed by ethanol precipitation and resuspension with TE buffer containing 50 mg/ml RNase A (#EN0531; Thermo Fisher Scientific). The obtained genomic DNA samples were subjected to PCR using primers indicated in the Supplemental Information. The cells were pelleted and boiled/vortexed for Western blotting analyses with 1X SDS–PAGE sample buffer. The samples were analyzed using antibodies as described in each figure legend.

Specific gene-targeted cell lines were developed using the *OsTIR1* parental DLD-1 line (Hassebroek et al, 2020). Targeting donor plasmids and pX330 guide plasmids were transfected to isolate candidate clones, which were screened by genomic PCR to confirm accurate transgene integration. The ability of Aux to deplete the protein was tested by Western blotting and immunostaining. Fluorescent fusion clones were verified by Western blotting and fluorescence microscopy. mCherry-PICH wild-type or mutant replacement cell lines were engineered using CRISPR/Cas9 in the PICH-AID cell line by inserting the rescue candidate gene at the *CCR5* locus. These clones were validated for transgene integration using genomic PCR analysis, and Western blot confirmed mCherry-fusion protein expression upon Dox addition. The H2B miRFP680, mNeon-Haspin, CENP-A-miRFP680, and mNeon-Bub1 lines were created using the OsTIR1/AID-PICH line and validated by genomic PCR and Western blot as shown in Fig S1.

## Preparation of the whole-cell lysate and mitotic chromosome isolation

The whole-cell lysate was prepared for testing AID-based degradation and Tet-On replacement system, cells were plated on the 5-cm dish, 0.5 mM auxin was added to allow degradation of the endogenous AID-tagged protein, and 1 μg/ml doxycycline was added for expressing the exogenous protein. After 10 h, cells were harvested and boiled/vortexed with 1X SDS–PAGE sample buffer and subjected to Western blot with indicated antibodies as described in figure legends. Mitotic shake-off was done to isolate mitotic chromosomes from the mitotic cells. For this, the cells were plated around 70% confluency and treated with 2 mM thymidine. After 18 h, the cells were released from the thymidine block by three washes of McCoy's media without FBS and placed in fresh media containing 7.5% FBS followed by the addition of 0.5 mM auxin and 1 μg/ml doxycycline as required. After 6 h, the cells were treated with 100 ng/ml nocodazole for an additional 4 h and mitotic cells were collected by doing a mitotic shake-off. The cells were then resuspended and lysed with lysis buffer (250 mM sucrose, 20 mM Hepes, 100 mM NaCl, 1.5 mM MgCl$_2$, 1 mM EDTA, 1 mM EGTA, 0.2% Triton X-100, 1:25 protease inhibitor [1 tablet dissolved in 2 ml water; MilliporeSigma], and 20 mM iodoacetamide [#I1149; Sigma-Aldrich]) and incubated for 5 min on ice. The lysed cells were then placed on a 40% glycerol-containing 0.25% Triton X-100 cushion and spun at 10,000*g* for 5 min twice. The isolated chromosomes were boiled and vortexed with 1X SDS–PAGE sample buffer and subjected to Western blotting.

## Western blotting

SDS–PAGE protein samples were loaded and separated on gradient gels (Invitrogen/Thermo Fisher Scientific). They were then transferred to a methanol-activated PVDF membrane with an ECL semi-dry transfer unit (Amersham Biosciences). After blocking with 5% casein, the proteins of interest were probed using primary antibodies as specified in each figure legend. Secondary antibodies (IRDye 800CW and IRDye 680 RD from LI-COR) were used for detection, and signals were visualized with the LI-COR Odyssey Fc machine.

## Live-cell imaging

For quantification of mitotic progression in live cells using miRFP680-H2B (Fig 1) and mCherry-TopoII (Fig 2), cells were plated on a 35-mm dish (Ibidi) at 60% confluency, and 2 mM thymidine was added followed by 0.5 mM auxin and 1 μg/ml doxycycline. Cells were released from the thymidine block after 18 h and Aux and Dox added back in imaging medium (Sigma-Aldrich). After 6 h, miRFP680 was imaged under normal cell culture conditions, at 37°C and 5% CO$_2$, using a Delta Vision Ultra microscope fitted with an Olympus U-Plan S-Apo 20X/0.75 or an Olympus 60X/1.42, Plan Apo N objective (UIS2, 1-U2B933) and a PCO-Edge sCMOS camera (>82% QE), using the following acquisition parameters. The images were taken every 5 min for 6 h. Entire mitotic cell volumes were obtained by capturing 16-μm-thick Z-series with 2-μm spacing. Z-series were then deconvolved and projected using SoftWoRx software, and the duration of mitosis was analyzed by looking at the miRFP680 signal, counting the time from chromosome condensation to anaphase onset using ImageJ software.

The same imaging conditions were used for imaging mNeon-Mad1 (Fig 3) and mNeon-Bub1 (Fig 5), except that after cells were plated on 35-mm dishes (Ibidi), asynchronous cells were imaged 24 h later. Imaging was done under normal cell culture conditions, at 37°C and 5% CO$_2$, using a Delta Vision Ultra microscope fitted with an Olympus 60X/1.42, Plan Apo N objective (UIS2, 1-U2B933) and a PCO-Edge sCMOS camera (>82% QE), using the following acquisition parameters. The images were taken every 5 min for 6 h. Entire mitotic cell volumes were obtained by capturing 24-μm-thick Z-series with 2-μm spacing. Z-series were then deconvolved and projected using SoftWoRx software. In the case of Mad1 foci, the time from NEBD to when the Mad1 foci disappeared was analyzed. In the case of Bub1 foci, the time from NEBD to anaphase onset was measured.

## Purification of His6-SUMO2–modified chromosomal proteins

For isolating SUMOylated proteins, the endogenous SUMO-2 protein was tagged with His6-3xFLAG as described above. Chromosome samples were collected as mentioned above, and lysis buffer (6 M guanidine HCl, 0.5% CHAPS, 30 mM Hepes, and 1 mM TCEP) was added and rotated overnight. Talon beads (#635502; Takara Bio) were treated with a 2.5% gelatin-blocking solution and rotated in a rotator overnight at RT. The samples were then sonicated, followed by centrifuging at 10,000*g* for 20 min, and the supernatant was mixed with the beads for binding overnight. After binding, beads

were washed with wash buffer 1 (6 M guanidine HCl, 10 mM Tris–HCl, pH 8.5, 300 mM NaCl, 1% Triton X-100, and 1 mM MgCl$_2$) followed by washing with wash buffer 2 (8 M urea, 10 mM Tris–HCl, pH 8.5, 300 mM NaCl, 1% Triton X-100, and 1 mM MgCl$_2$). After washing, the beads were sent for mass spectrometry analysis.

### Morphological quantification of Haspin distribution with wndchrm

The wndchrm analysis was done as described previously (Sundararajan et al, 2023). Images for wndchrm analyses were acquired from live cells expressing fluorescent Haspin, as described above. For quantification of morphological similarities/ dissimilarities, a supervised machine learning algorithm, wndchrm (weighted neighbor distance using a compound hierarchy of algorithms representing morphology), was used, as previously described. Classification accuracies were obtained by comparison of the indicated number of images between the control and subjects by the wndchrm algorithm.

### Proteomics sample preparation and analysis

For on-bead digestion, we modified the protocol from Branon et al (2018). Beads were trypsinized to digest the interacting proteins. Beads were washed and digested in 2 M urea, 50 mM Tris–HCl, pH 8.5, and trypsin. Digestion was performed for 60 m at 25°C in a thermomixer, shaking at 800 rpm. After the initial digestion, the supernatant was removed and collected into freshly labeled tubes. Beads were washed twice in buffer (containing 2 M urea, 50 mM Tris–HCl, pH 8.5, and 100 mM TCEP), and the supernatants were pooled. Samples were digested overnight at 25°C in a thermomixer, shaking at 800 rpm. Samples were desalted on C18 StageTips and evaporated to dryness in a vacuum concentrator.

All peptide digests were analyzed using reversed-phase (RP) chromatography on an UltiMate 3000 RSLCnano system coupled with a Q Exactive HF mass spectrometer. Enriched peptide pellets were dissolved in 2% ACN/0.1% formic acid, and a relatively equal amount of peptide digests from each sample were loaded onto an Acclaim PepMap 100 trapping cartridge using the loading pump flowing at 5 µl/min. Trapped peptides were eluted onto a 75 $\mu$m I.D. × 15 cm length C18 column packed in-house with 1.9 $\mu$m ReproSil-Pur 120 C18-AQ (Dr. Maisch GmbH). The mobile phases were 0.1% formic acid (buffer A) and 80% acetonitrile and 0.1% formic acid (buffer B). Peptides were separated using a 130-min RP gradient (wash with 2% buffer B for 5 min, 2–25% in 82 min, 25–40% in 20 min, 40–95% in 1 min, hold at 95% buffer B for 10 min, and wash with 2% buffer B for 12 min) at a flow rate of 300 nl/min. Peptides were eluted directly from the tip of the column and nano-sprayed into the mass spectrometer.

The Q Exactive HF mass spectrometer was operated in the data-dependent MS/MS mode, using Xcalibur software (Thermo Fisher Scientific). The spray voltage was set at 2.5 kV, and the heated capillary temperature was set at 275°C. Peptides were fragmented by higher energy collisional dissociation (HCD) in positive polarity mode with normalized collision energy of 27. The survey scans were acquired at a resolution of 60,000 at m/z 200, and the mass range was set to m/z 375–1,600. The top 15 most abundant ions with a charge ranging from 2 to 5 were selected from the survey scans for fragmentation, the fragment scan resolution was set to 15,000, and the isolation window was set to 1.4. Dynamic exclusion was employed for the 20 s.

RAW data files were processed using Proteome Discoverer 2.5 (PD2.5). The data files were searched against a *Homo sapiens* (*Hs*) database and a contaminant database. The *Hs* database consists of 89832 non-redundant proteins (downloaded from NCBI on 2023-10-11). The contaminant database consists of 426 usual contaminants (such as human keratins, IgGs, and proteolytic enzymes). The data were searched using the SEQUEST algorithm in PD2.5. Oxidation of methionine residues (+15.995 Da) and deamidation of asparagine, glutamine, and arginine (+0.984 Da) were set as a variable modification; carbamidomethylation of cysteine residues (+57.021 Da) was set as static modifications. In addition to that, N-terminal glutamate-to-pyroglutamate conversion (−17.027 D) on the peptide terminus and acetylation (+42.011) at the N terminus of protein were set as variable modifications. The peptide-spectrum matches (PSMs) were adjusted to a 1% false discovery rate (FDR). In addition to that, the minimum number of peptide-spectrum matches (PSMs) was set to 2. Data were analyzed, and results were generated in Excel files. NSAFs were calculated for each detected protein, as described previously (Zybailov et al, 2005).

### Quantification and statistical analysis

Quantifiable data were analyzed for statistical significance using GraphPad Prism software (version 9). Depending on the data, either a one-way ANOVA or a paired two-tailed *t* test was used, followed by appropriate post hoc tests where necessary. Detailed information about the statistical analysis is provided in the figure legends.

# Supplementary Information

# Acknowledgements

This work was initially supported by NIH/NIGMS, GM112793, GM130858, and P20GM103418, then currently by NIH/NCI R21CA259718. The following funds from the University of Kansas were used to support student salaries and for mass spectrometry analysis: KUCC/CB pilot grant (KAN1000623) and general research funds from the University of Kansas (#2144083). Image analysis by machine learning program part was supported by JSPS KAKENHI Grant Numbers JP18H05531, JP18K19310, and JP20H03520 [to N Saitoh] and by grants from the Vehicle Racing Commemorative Foundation [to N Saitoh]. V Aksenova, A Arnaoutov, and M Dasso are supported by NIH/NICHD Intramural projects Z01 HD008954 and ZIA HD001902. Fig 5F was created in BioRender. B Lama (2025) https://BioRender.com/z47x712.

### Author Contributions

B Lama: conceptualization, data curation, formal analysis, validation, investigation, visualization, and writing—original draft.
H Park: resources and data curation.

A Saraf: data curation, formal analysis, validation, and methodology.

V Hassebroek: resources, data curation, and methodology.

D Keifenheim: data curation and methodology.

T Saito-Fujita: formal analysis, visualization, and methodology.

N Saitoh: formal analysis, validation, funding acquisition, and methodology.

V Aksenova: resources and methodology.

A Arnaoutov: resources and methodology.

M Dasso: resources, funding acquisition, and methodology.

DJ Clarke: conceptualization, data curation, formal analysis, validation, funding acquisition, investigation, and writing—review and editing.

Y Azuma: conceptualization, resources, supervision, funding acquisition, investigation, methodology, and writing—review and editing.

## Conflict of Interest Statement

The authors declare that they have no conflict of interest.

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
