## [Reviewer comments · Life Science Alliance]

Life Science Alliance

PICH impacts the Spindle Assembly Checkpoint via its DNA translocase and SUMO-interaction activities

Bunu Lama, Hyewon Park, Anita Saraf, Victria Hassebroek, Daniel Keifenheim, Tomoko Saito-Fujita, Noriko Saitoh, Vasilisa Aksenova, Alexei Arnaoutov, Mary Dasso, Duncan Clarke, and Yoshiaki Azuma

DOI: <https://doi.org/10.26508/lsa.202403140>

Corresponding author(s): Yoshiaki Azuma, University of Kansas

Review Timeline:

Submission Date:	2024-11-19
Editorial Decision:	2024-11-20
Revision Received:	2025-01-27
Editorial Decision:	2025-01-28
Revision Received:	2025-01-30
Accepted:	2025-01-31

Transaction Report:

Please note that the manuscript was previously reviewed at another journal and the reports were taken into account in the decision-making process at *Life Science Alliance*.

Reviewer #1

Comments to the Authors (Required):

In this manuscript, Lama and colleagues argue that PICH remodels SUMOylated proteins to ensure correct temporal silencing of the spindle assembly checkpoint. The main observation in support of this idea is that depletion of PICH, or re-expression in cells lacking endogenous PICH of exogenous PICH mutants lacking SUMO binding ability or ATPase activity (respectively identified as PICH Δ 3SIM and K128A) are (very slightly) delayed in mitosis. The authors asked if this short arrest was caused by activation of a Topo2alpha-dependent pathway (characterized in a previous paper and named TRC, for Topo2alpha responsive checkpoint). After concluding that this is not the case, they turned to the spindle assembly checkpoint (SAC) and found prolonged retention of the checkpoint protein MAD1 on kinetochores upon PICH depletion or in cells expressing dysfunctional PICH mutants. Because PICH is known to interact with SUMOylated proteins, the authors hypothesized that PICH depletion, or its replacement with mutants, may lead to the accumulation of SUMOylated proteins, and that this may be the cause of the observed mitotic delay. To test this idea, the authors generated a cell line expressing a tagged version of SUMO2, and compared the abundance of SUMO2-conjugated proteins in presence or absence of PICH function. This identified several proteins whose SUMOylation appeared to increase when PICH function had been impaired. Among these, the authors identified BUB1, and demonstrated that BUB1 kinetochore levels were slightly increased upon depletion of PICH, and that this effect may be due to a defect in recovery from checkpoint activation. The authors' model is that PICH facilitates the removal of SUMOylated proteins from the kinetochore to promote checkpoint silencing.

The work presented here was enabled by the creation of several cell lines, and clearly reflects a considerable and valuable effort by the authors. The main limitation of the study is that the observed effects are very minor, and that there is no ultimate evidence that the function of PICH that causes these effects is exquisitely and exclusively regulatory. It may rather reflect small persisting attachment errors, maybe caused by small problems in the organization of centromeric chromatin, that signal to the SAC. I.e. the delay may not reflect a mere silencing error but persistent checkpoint activation, an issue that the authors do not address and that would be very difficult to address, given the small entity of the arrest. In this regard, also the proposed model that identifies excessive SUMOylation as the cause of the mitotic delay, while not implausible, comes across as poorly supported at this stage of the analysis. Increased SUMOylation was observed in the absence of PICH, but cells were able to leave mitosis only a few minutes after the control cells, implying that other proteins processing the excess of SUMO must be present. As the authors have not excluded that the mitotic delay is merely caused by a genuine SAC activation, the role of PICH in the control of SUMOylation remains uncertain. Collectively, I feel therefore that study, albeit valuable, does not yet represent a fully compelling conceptual or mechanistic advancement.

Other concerns

- The differences of mitotic timing in Δ PICH cells in Figure 1c and 2c raises a concern of consistency. Why is the timing of mitotic exit different in these two conditions?
- In Figure 3, the persistence of MAD1 at kinetochores in Δ PICH cells extends well over 50 minutes, i.e. considerably more than it is required for these cells to exit mitosis (~35 minutes, as evinced from Figure 1). This seems rather implausible, as the loss of MAD1 from kinetochores invariably precedes mitotic exit.

Minor points

- Figure 1B: last row, 5th panel, partly hidden text on the bottom-right
- Figure 1C: It would be helpful if the authors indicated the mean time of mitotic exit for the various conditions shown in this graph.
- Indicate in text and relevant figure that Topo2alpha is FLAG-tagged

Reviewer #2

Comments to the Authors (Required):

This paper explores the function of PICH during mitosis using engineered cell lines that allows the authors to deplete PICH and reconstitute with mutants and monitor effects through time-lapse microscopy. The authors observe that in their system the depletion of PICH causes a mitotic delay which correlates with prolonged Mad1 and Bub1 on kinetochores. The authors identify a number of proteins, the SUMOylation of which, might be altered by the absence of PICH and an inactive version of PICH. The authors propose that SUMOylation of Bub1 might be the cause of the prolonged SAC in PICH depleted cells.

Although the cellular assays are nicely conducted this is a very descriptive work that lacks the mechanistic insights needed for publication in this journal. At present this represents some interesting observations, but they cannot be linked directly and therefore claims are not strongly supported. Without showing that SUMO2/3 modification of Bub1, in a PICH regulated manner, is the cause of prolonged SAC then the observed phenotypes can be caused by many indirect effects on the SAC.

Therefore, I do not support publication.

A few comments:

- 1) I think it would help the reader if there was a schematic of PICH in figure 1 and also an outline of the experimental setup and the kinetics of PICH depletion. Is PICH depletion occurring just before cells enter mitosis? The western blots is from 22hrs of treatment and if this is also the case for filming then I do not understand the effort in generating AID tagged cell lines. The purpose of AID is to acutely deplete and look at immediate phenotypes to avoid indirect long term effects.
- 2) Figure 2 - without the conditions where PICH is present it is difficult to judge these results. Why is a different statistical method used compared to Figure 1?
- 3) As they show in figure 1 that chromosomes take longer to align this figure is almost trivial as Mad1 will off course respond to this. This does not mean that PICH has anything to do with the SAC.
- 4) I think the rational in Figure 4 is difficult to follow. It assumes that PICH always extracts SUMO2/3 modified proteins from cells. I would think that the PICH SIM mutant would have been a useful control here. Also I lack a more complete description of the MS results - total identified proteins per condition, overlap of upregulated AND downregulated proteins.
- 5) Figure 5 has no link to figure 4 and is just recapitulating observations in figure 1 and 3.

November 20, 2024

Re: Life Science Alliance manuscript #LSA-2024-03140-T

Dr. Yoshiaki Azuma
University of Kansas
Dept. of Molecular Biosciences
1049 Haworth Hall
Haworth Hall Rm.3037
Lawrence, Kansas 66045

Dear Dr. Azuma,

Thank you for submitting your manuscript entitled "PICH controls the Spindle Assembly Checkpoint by its DNA translocase and SUMO-interaction activities" to Life Science Alliance. We invite you to submit a revised manuscript addressing the Reviewer comments.

Thank you for this interesting contribution to Life Science Alliance. We are looking forward to receiving your revised manuscript.

Sincerely,

B. MANUSCRIPT ORGANIZATION AND FORMATTING:

Point by point response to critiques.

Reviewer #1 (Comments to the Authors (Required)):

In this manuscript, Lama and colleagues argue that PICH remodels SUMOylated proteins to ensure correct temporal silencing of the spindle assembly checkpoint. The main observation in support of this idea is that depletion of PICH, or re-expression in cells lacking endogenous PICH of exogenous PICH mutants lacking SUMO binding ability or ATPase activity (respectively identified as PICH Δ 3SIM and K128A) are (very slightly) delayed in mitosis. The authors asked if this short arrest was caused by activation of a Topo2alpha-dependent pathway (characterized in a previous paper and named TRC, for Topo2alpha responsive checkpoint). After concluding that this is not the case, they turned to the spindle assembly checkpoint (SAC) and found prolonged retention of the checkpoint protein MAD1 on kinetochores upon PICH depletion or in cells expressing dysfunctional PICH mutants. Because PICH is known to interact with SUMOylated proteins, the authors hypothesized that PICH depletion, or its replacement with mutants, may lead to the accumulation of SUMOylated proteins, and that this may be the cause of the observed mitotic delay. To test this idea, the authors generated a cell line expressing a tagged version of SUMO2, and compared the abundance of SUMO2-conjugated proteins in presence or absence of PICH function. This identified several proteins whose SUMOylation appeared to increase when PICH function had been impaired. Among these, the authors identified BUB1, and demonstrated that BUB1 kinetochore levels were slightly increased upon depletion of PICH, and that this effect may be due to a defect in recovery from checkpoint activation. The authors' model is that PICH facilitates the removal of SUMOylated proteins from the kinetochore to promote checkpoint silencing.

The work presented here was enabled by the creation of several cell lines, and clearly reflects a considerable and valuable effort by the authors. The main limitation of the study is that the observed effects are very minor, and that there is no ultimate evidence that the function of PICH that causes these effects is exquisitely and exclusively regulatory. It may rather reflect small persisting attachment errors, maybe caused by small problems in the organization of centromeric chromatin, that signal to the SAC. I.e. the delay may not reflect a mere silencing error but persistent checkpoint activation, an issue that the authors do not address and that would be very difficult to address, given the small entity of the arrest. In this regard, also the proposed model that identifies excessive SUMOylation as the cause of the mitotic delay, while not implausible, comes across as poorly supported at this stage of the analysis. Increased SUMOylation was observed in the absence of PICH, but cells were able to leave mitosis only a few minutes after the control cells, implying that other proteins processing the excess of SUMO must be present. As the authors have not excluded that the mitotic delay is merely caused by a genuine SAC activation, the role of PICH in the control of SUMOylation remains uncertain. Collectively, I feel therefore that study, albeit valuable, does not yet represent a fully compelling conceptual or mechanistic advancement.

We agree that we cannot propose a definitive molecular model and, as this reviewer indicated, indirect effects on the mitotic machinery, via centromere or kinetochore structural problems, could account for the observed persistent activation of the SAC. Against the possibility that centromeric chromatin or kinetochores are broadly affected in the absence of PICH, we do not observe a change in CENP-A abundance, which we include in figure 5c. However, we note that CENP-B abundance was affected by loss of PICH and changes in this key centromeric protein could account for SAC activation via attachment errors as proposed by the reviewer.

In Summary, because loss of PICH changed the abundance of both SAC regulatory proteins and centromere/kinetochore proteins, the data do not distinguish between direct and indirect SAC activation. Nevertheless, the reviewer stated our data are valuable and our study will clearly promote further mechanistic investigation of the role of PICH-dependent SUMOylated protein remodeling that is known to be important for faithful mitosis.

Other concerns

-The differences of mitotic timing in Δ PICH cells in Figure 1c and 2c raises a concern of consistency. Why is the timing of mitotic exit different in these two conditions?

The reviewer raises an important question. Indeed, the time from NEBD (nuclear envelope breakdown) to anaphase was not consistent between the experiments in Fig 1c (~40 minutes) and 2c (~50 minutes). These differences are likely due to the fact that the experiments used different cell lines where H2B was tagged in Fig 1c and TopoII was tagged in Fig 2c. There may be slight differences in mitotic timing caused directly by the tags introduced. In addition, the experimental conditions differ between these experiments because Fig 2c required endogenous TopoII depletion and induction of exogenous tagged TopoII, which was not the case for the experiments in Fig 1c. Nevertheless, among the different experiments using the same cell lines and conditions, there was consistency, and the statistics support the conclusions we reach.

-In Figure 3, the persistence of MAD1 at kinetochores in Δ PICH cells extends well over 50 minutes, i.e. considerably more than it is required for these cells to exit mitosis (~35 minutes, as evinced from Figure 1). This seems rather implausible, as the loss of MAD1 from kinetochores invariably precedes mitotic exit.

We agree with the reviewer that the time from NEBD to loss of Mad1 from kinetochores (Fig 3) ought to be shorter than the time from NEBD to anaphase (Fig 1 and 2). This discrepancy is likely due to the fact that we measured different signals, i.e. mNEON-Mad1 versus miRFP680-H2B or mCherry-TopoII. These differences likely affect the exact starting point (NEBD) and end point of our measurements. The exact timing of NEBD is precise when observing Mad1 because it allows direct visualization of nuclear pore disassembly. On the other hand, the TopoII and H2B signals only allow monitoring of partially condensed chromosome positions/behavior as a proxy for NEBD and this is likely a less accurate parameter. We have discussed these technical factors in the revised manuscript. As discussed in the point above, clonal variation between the

cell lines might be an additional factor affecting the exact timing since the cell lines used are different. Perhaps a more impactful factor is that tagging Mad1 could affect the timing of mitotic progression, being a key SAC regulator. Regardless of these technical factors, we would like to emphasize that among the different experiments using the same cell lines and conditions, there was consistency, and the statistics support the conclusions we reach.

Minor points

- Figure 1B: last row, 5th panel, partly hidden text on the bottom-right : **We thank the reviewer for pointing this out and we have fixed this in the revised manuscript.**
- Figure 1C: It would be helpful if the authors indicated the mean time of mitotic exit for the various conditions shown in this graph. : **We have indicated this in the revised manuscript.**
- Indicate in text and relevant figure that Topollalpha is FLAG-tagged : **We have fixed this in the revised manuscript.**

Reviewer #2 (Comments to the Authors (Required)):

This paper explores the function of PICH during mitosis using engineered cell lines that allows the authors to deplete PICH and reconstitute with mutants and monitor effects through time-lapse microscopy. The authors observe that in their system the depletion of PICH causes a mitotic delay which correlates with prolonged Mad1 and Bub1 on kinetochores. The authors identify a number of proteins, the SUMOylation of which, might be altered by the absence of PICH and an inactive version of PICH. The authors propose that SUMOylation of Bub1 might be the cause of the prolonged SAC in PICH depleted cells.

Although the cellular assays are nicely conducted this is a very descriptive work that lacks the mechanistic insights needed for publication in this journal. At present this represents some interesting observations, but they cannot be linked directly and therefore claims are not strongly supported. Without showing that SUMO2/3 modification of Bub1, in a PICH regulated manner, is the cause of prolonged SAC then the observed phenotypes can be caused by many indirect effects on the SAC.

Therefore, I do not support publication.

We thank the reviewer for acknowledging the power of our cellular assays, for elaborating that the experiments were technically conducted well, and that our study revealed interesting observations. We agree that without detecting SUMOylated Bub1 and its behavior after loss of PICH function, the molecular details of our model might not be supported strongly. In the revised manuscript we have made it clear that the “PICH-remodeling SUMOylated Bub1 model” is not the only possibility to explain our data and we emphasize the potential alternative mechanisms.

Accordingly, we have also expanded the interpretation/description of the MassSpec data identifying PICH targets as indicated below.

A few comments:

1) I think it would help the reader if there was a schematic of PICH in figure 1 and also an outline of the experimental setup and the kinetics of PICH depletion. Is PICH depletion occurring just before cells enter mitosis? The western blots is from 22hrs of treatment and if this is also the case for filming then I do not understand the effort in generating AID tagged cell lines. The purpose of AID is to acutely deplete and look at immediate phenotypes to avoid indirect long term effects.

We have added schematics of experiment in each figure in the revised MS and have indicated that longer treatment was needed for expression of exogenous alleles, not depletion of PICH. Depletion can be achieved within 6 hours but Dox-induction for detectable levels of exogenous PICH and TopoII required longer incubation in these cell lines. We explained that issue in the results section by including the statement, as such "*To obtain optimal PICH expression and number of mitotic cells, we synchronized cells by single thymidine arrest/release procedure with Aux/Dox treatment as indicated in Fig 1C*".

2) Figure 2 - without the conditions where PICH is present it is difficult to judge these results. Why is a different statistical method used compared to Figure 1?

We have now provided data for TopoII α -wt and TopoII α -3KR replaced cell lines as shown in Supplemental Figure 2 A (without PICH depletion). We have also made it clear in the revised manuscript that these cell lines have differences in genetic background from those in Figure 2, and so direct comparisons could be limited. Nevertheless, at least, we can say that the 3KR mutations do not increase duration of mitosis.

As for the different statistical methods used, this is because of the sample number differences: figure 1 has five conditions but figure 2 only has two conditions, so we could not use one way ANOVA for figure 2.

3) As they show in figure 1 that chromosomes take longer to align this figure is almost trivial as Mad1 will of course respond to this. This does not mean that PICH has anything to do with the SAC.

We agree with the reviewer that we have not distinguished a direct role for PICH in SAC regulation versus an indirect effect of PICH depletion that affects chromosome alignment. To make this clear, we have slightly changed the manuscript title so that it is not misleading: from "PICH controls SAC" to "PICH impacts SAC". We also revised the discussion to make it clearer we are not claiming exclusively that PICH directly controls the SAC. We also clarified

that the data in Fig 1 do not measure the duration of chromosome alignment but rather measure the time from NEBD to anaphase onset.

4) I think the rationale in Figure 4 is difficult to follow. It assumes that PICH always extracts SUMO2/3 modified proteins from cells. I would think that the PICH SIM mutant would have been a useful control here. Also I lack a more complete description of the MS results - total identified proteins per condition, overlap of upregulated AND downregulated proteins.

In the revised manuscript we have added a heat map of repeatedly detected known chromosomal proteins (See Supplemental Figure 3) and we have included this information in the manuscript text. We also discussed proteins with reduced abundance on chromosomes. However, we stress that our proteomics analysis enriched/isolated SUMOylated proteins and detected their relative abundance in the chromosomal fraction. Based on the known function of PICH as a SUMOylated chromosome protein remodeling DNA translocase, it is hard to rationalize direct effects of PICH loss-of-function that would lead to reduced abundance of a SUMOylated protein on chromosomes. Indeed, we clearly observed a dramatic overall increase in SUMOylated chromosome proteins under these conditions. We do agree that there could be indirect effects of PICH loss-of-function that would decrease the abundance of a SUMOylated chromosome protein. For example, PICH could remodel a chromosomal protein such that it provides optimal conditions for that protein to be SUMOylated making its level low when PICH is depleted. We have hesitation to discuss such explanations in detail because we currently do not have any supportive evidence from our experiments (other than the relative abundance measurement from this MassSpec), and so we would be speculating.

5) Figure 5 has no link to figure 4 and is just recapitulating observations in figure 1 and 3.

We agree with the reviewer that the importance of Fig 5 a,b was not made clear. In the revised manuscript we have made it clear that Fig 5 a,b present important data because these experiments validate the mass spec data in Fig 4. Also, the results suggest increasing Bub1 abundance specifically at centromeres/kinetochores could activate Mad1 for prolonged SAC engagement. This spatial analysis of Bub1 abundance extends the bulk abundance analysis we gain from the Mass spec. We agree with the reviewer that the data in Fig 5 a,b are predicted based on the mass spec data, but Fig 5 provides clear confirmation that loss of PICH enhances Bub1 association with centromeres/kinetochores which is predicted to impact the SAC through the Bub1/Mad1 pathway.

January 28, 2025

RE: Life Science Alliance Manuscript #LSA-2024-03140-TR

Dr. Yoshiaki Azuma
University of Kansas
Dept. of Molecular Biosciences
1049 Haworth Hall
Haworth Hall Rm.3037
Lawrence, Kansas 66045

Dear Dr. Azuma,

Thank you for submitting your revised manuscript entitled "PICH impacts the Spindle Assembly Checkpoint via its DNA translocase and SUMO-interaction activities". We would be happy to publish your paper in Life Science Alliance pending final revisions necessary to meet our formatting guidelines.

- please be sure that the authorship listing and order is correct
- please add the Twitter/X and Bluesky handles of your host institute/organization as well as your own or/and one of the authors in our system
- please remove track changes from the manuscript file
- please consult our manuscript preparation guidelines <https://www.life-science-alliance.org/manuscript-prep> and make sure your manuscript sections are in the correct order
- please use the [10 author names, et al.] format in your references (i.e. limit the author names to the first 10)
- please add a callout for Figure 4E to your main manuscript text

A. FINAL FILES:

B. MANUSCRIPT ORGANIZATION AND FORMATTING:

Thank you for your attention to these final processing requirements. Please revise and format the manuscript and upload materials within 5 days.

Sincerely,

January 31, 2025

RE: Life Science Alliance Manuscript #LSA-2024-03140-TRR

Dr. Yoshiaki Azuma
University of Kansas
Dept. of Molecular Biosciences
1049 Haworth Hall
Haworth Hall Rm.3037
Lawrence, Kansas 66045

Dear Dr. Azuma,

Thank you for submitting your Research Article entitled "PICH impacts the Spindle Assembly Checkpoint via its DNA translocase and SUMO-interaction activities". It is a pleasure to let you know that your manuscript is now accepted for publication in Life Science Alliance. Congratulations on this interesting work.

DISTRIBUTION OF MATERIALS:

Again, congratulations on a very nice paper. I hope you found the review process to be constructive and are pleased with how the manuscript was handled editorially. We look forward to future exciting submissions from your lab.

Sincerely,
